# Aromatic Polyimide Membranes with *tert*-Butyl and Carboxylic Side Groups for Gas Separation Applications—Covalent Crosslinking Study

**DOI:** 10.3390/polym14245517

**Published:** 2022-12-16

**Authors:** Noelia Esteban, Marta Juan-y-Seva, Carla Aguilar-Lugo, Jesús A. Miguel, Claudia Staudt, José G. de la Campa, Cristina Álvarez, Ángel E. Lozano

**Affiliations:** 1IU CINQUIMA, University of Valladolid, Paseo Belén 5, E-47011 Valladolid, Spain; 2Institute of Polymer Science and Technology, ICTP-CSIC, Juan de la Cierva 3, E-28006 Madrid, Spain; 3Facultad de Química, Universidad Nacional Autónoma de México, Cd. Universitaria, Coyoacán, Mexico City 04510, Mexico; 4Global Product and System Development Personal Care, BASF, 40789 Monheim, Germany; 5SMAP, UA-UVA_CSIC, Associated Research Unit to CSIC, University of Valladolid, Paseo Belén 7, E-47011 Valladolid, Spain

**Keywords:** aromatic polyimides, crosslinking, plasticization resistance, gas separation

## Abstract

A set of aromatic copolyimides was obtained by reaction of 4,4′-(hexafluoroisopropylidene) diphthalic anhydride (6FDA), and mixtures of the diamines 1,4-bis(4-amino-2-trifluoromethylphenoxy)-2,5-di-*tert*-butylbenzene (CF_3_TBAPB) and 3,5-diamino benzoic acid (DABA). These polymers were characterized and compared with the homopolymer derived from 6FDA and CF_3_TBAPB. All copolyimides showed high molecular weight values and good mechanical properties. The presence of carboxylic groups in these copolymers allowed their chemical crosslinking by reaction with 1,4-butanediol. Glass transition temperatures (Tg) were higher than 260 °C, showing the non-crosslinked copolyimides had the highest Tg values. Degradation temperature of crosslinked copolyimides was lower than their corresponding non-crosslinked ones. Mechanical properties of all polymers were good, and thus, copolyimide (precursor, and crosslinked ones) films could be tested as gas separation membranes. It was observed that CO_2_ permeability values were around 100 barrer. Finally, the plasticization resistance of the crosslinked material having a large number of carboxylic groups was excellent.

## 1. Introduction

Currently there are a large number of glassy polymeric membranes that show good productivity for gas separation at high pressures [1,2], but they exhibit some drawbacks due to physical aging processes, and to their tendency to undergo plasticization processes as result of passing condensable gases through the membrane [3,4,5,6,7].

The plasticization occurs when the membrane is exposed to highly soluble gases, such as carbon dioxide and small hydrocarbons (C ≤ 3), which produce significant swelling of the polymeric matrix, increasing both the polymer’s free volume and the chains’ segmental mobilities [8,9]. As a consequence of this process, the permeability of the membrane increases while its selectivity decreases, which leads to a very marked deterioration of its properties as a gas separation material [10].

Plasticization produces severe problems in applying these materials at the industrial level. Thus, for example, natural gas consists mainly of CH_4_, variable fractions of heavy gaseous hydrocarbons (ethane, propane, butane, among others), gases such as N_2_ and CO_2_, and traces of many other aromatic and paraffinic compounds. Among these components, CO_2_, ethane, propane, and butane have a very high capacity to plasticize the membranes [11,12,13,14,15,16,17], preventing the application of polymeric membranes in the high-pressure separation of CO_2_ and hydrocarbons for natural gas purification [17,18]. Plasticization also occurs in separations that involve other condensable gases of high economic importance [15], such as ethene/ethane [16], propylene/propane [17,18], or butadiene/butane [19].

In particular, the CO_2_-induced plasticization results in a permeability minimum at a specific pressure (plasticization pressure, Figure 1), with a time dependence of CO_2_ permeability above the plasticization pressure and a significant decrease of CO_2_ selectivity at elevated pressures [19,20,21,22]. Additionally, plasticization depends on other aspects that affect the gas separation properties: time [8,23], pressure [24], temperature [25,26], and thickness of the tested films [27,28].

One of the most commonly used methods to avoid or minimize the plasticization process consists of crosslinking the polymer chains, which reduces their mobility and restricts the volume increase in the membrane due to the presence of condensable gases [29,30,31,32,33,34,35]. Furthermore, crosslinking produces improvements in the chemical stability of polymers [36,37], as has been described not only in gas separation processes [29,38,39] but also in other separation processes, such as pervaporation [40,41]. However, plasticization can also occur in crosslinked polymers, although it takes place at higher pressures or longer exposure times. Nevertheless, it is important to note that crosslinking is often accompanied by a decrease in permeability, which implies a decrease in productivity [4,15,42,43,44,45].

Covalent crosslinking is the most commonly used methodology. To carry out the crosslinking, di- or multifunctional nucleophilic reagents react with the electrophilic groups present in the polymer. Numerous works have been published in this sense, using diamines [46,47,48] or diols [24,42] as crosslinking agents. The difunctional crosslinking compounds act as spacer moieties between the chains, which prevent the permeability from decreasing as occurs in other methodologies [15,48].

Many of the works cited above are based on the study of copolymers derived from the 4,4′-(hexafluoroisopropylidene)diphtalic anhydride (6FDA), which have very interesting gas separation properties. However, it must be taken into account that they are very susceptible to plasticization at high working pressures [49,50].

Likewise, the copolyimides derived from 3,5-diamino benzoic acid (DABA) have been widely studied due to the remarkable capacity of functionalization provided by the high polarity carboxylic group (-COOH) [31,51,52,53,54,55,56].

In this research, the synthesis of copolymers formed by the reaction between the 6FDA dianhydride and mixtures of the diamines; 1,4-bis(4-amino-2-trifluoromethylphenoxy)-2,5-di-tert-butylbenzene (CF_3_TBAPB) and DABA, has been carried out with the aim of obtaining gas separation membranes with high plasticization resistance.

## 2. Materials and Methods

Anhydrous 1-methyl-2-pyrrolidone (NMP, Sigma-Aldrich, Madrid, Spain), anhydrous pyridine (Py, Sigma-Aldrich, Madrid, Spain), 4-dimethylaminopyridine (DMAP, Sigma-Aldrich, Madrid, Spain), trimethylchlorosilane (TMSCl, Sigma-Aldrich, Madrid, Spain), *p*-toluenesulfonic acid (Sigma-Aldrich, Madrid, Spain), 1,4-butanediol (Sigma-Aldrich, Madrid, Spain), acetic anhydride (Sigma-Aldrich, Madrid, Spain), and other commercial solvents (chloroform, CHCl_3_; *N*,*N*-dimethylacetamide; DMAc, tetrahydrofuran, THF, m-cresol) were obtained from commercial sources at their highest level of purity and were used as received. Acetic anhydride (Sigma-Aldrich, Madrid, Spain) was distilled before use.

3,5-diaminobenzoic acid (DABA, >97.5%, Sigma-Aldrich, Madrid, Spain) and 4,4′-(hexafluoroisopropylidene)diphtalic anhydride (6FDA, >98%, TCI, Tokyo, Japan) were purified by sublimation.

1,4-Bis(4-aminophenoxy)2,5-di-tert-butylbenzene (TBAPB) was synthesized according to previously reported methods [57,58,59], from 2,5-di-tert-butylhydroquinone (99%, Sigma-Aldrich, Madrid, Spain) and *p*-chloronitrobenzene (99%, Sigma-Aldrich, Madrid, Spain) via nucleophilic aromatic substitution in the presence of potassium carbonate (Panreac, Barcelona, Spain) and DMF (99.5%, Sigma-Aldrich, Madrid, Spain) as the solvent, followed by catalytic reduction with hydrazine hydrate (98%, Sigma-Aldrich, Madrid, Spain) and Pd/C (10% Pd, Sigma-Aldrich, Madrid, Spain) as the catalyst.

The 6FDA dianhydride was heated at 190 °C for 2 h just before use. Polymerizations were carried out in DMAc, using acetic anhydride and pyridine for chemical imidization of the polyamic acid precursor.

### 2.1. Characterization Methods

Monomers, polyimides, and copolyimides were characterized by attenuated total reflection Fourier transform infrared (ATR-FTIR), and proton nuclear magnetic resonance (^1^H NMR). ^1^H-NMR experiments were realized on a Bruker Avance 400 MHz spectrometer using DMSO-*d*_6_ as solvent. ATR-FTIR spectra were recorded on a Perkin Elmer RX-1 spectrometer with an ATR accessory.

Thermogravimetric analyses (TGA) were performed on a TA-Q500 analyzer using a Hi-Res^®^ method. The samples were heated to 20 °C/min in the temperature range from 30 to 850 °C. Nitrogen was used as the purge gas (60 mL/min).

Wide-angle X-ray scattering patterns (WAXS) were recorded on a Bruker D8 Advance equipped with a Goebel mirror and a Vantec PSD, with CuK radiation (wavelength, λ = 1.54 Å) in the scattering angle range from 3 to 60°.

Solubility tests were done in a test tube using 1–2 mg of polymer and 1 mL of the selected solvent, and placed under magnetic stirring for 24 h. After that, if the polymer was not soluble, it was heated at the solvent’s boiling point.

The density (ρ) of membranes was determined using a top-loading electronic XS105 Dual Range Mettler Toledo balance coupled with a density kit based on Archimedes’ principle. The samples were sequentially weighed in air and into high-purity isooctane at room temperature. The density was calculated from Equation (1).
(1)$$\rho =\rho_{\mathrm{liquid}}\times \frac{w_{\mathrm{air}}-w_{\mathrm{liquid}}}{w_{\mathrm{air}}}$$
where ρ_liquid_ is the density of isooctane, w_air_ is the weight of the sample in air, and w_liquid_ is its weight when submerged in isooctane. Four density measurements were made for each sample.

The fractional free volume (FFV) of membranes was calculated from their density data using the following Equation (2):(2)$$\mathrm{FFV}=\frac{V_{e}-1.3V_{w}}{V_{e}}$$
where V_e_ is the specific volume of the membrane and V_w_ is the polymer’s van der Waals volume that was calculated from the Biovia Materials Studio program [60]. A 20-unit polymer structure was built using the Polymer Builder and the Atom Volumes and Surface algorithms of the Materials Studio program. The polymer structure was optimized by using the Compass force field.

The crosslinking of the copolymers was checked by determining their gel fractions. The gel fraction, Gf(%), of film samples was estimated using Equation (3):(3)$$\mathrm{Gf}\left(\%\right)=100\times \frac{W_{\mathrm{final}}}{W_{\mathrm{inicial}}}$$
where W_initial_ is the weight of crosslinked sample and W_final_ is the weight of the crosslinked sample after extracting the soluble fraction. W_final_ was determined by immersing the copolymer samples in DMAc at room temperature for 48 h and at 60 °C for 1 h under stirring. Then, the sample was washed four times with ethanol/water (1/1) and dried at 60 °C for 4 h and at 120 °C under vacuum for 8 h and weighed.

Mechanical properties were measured using an MTS Synergie-200 testing machine equipped with a 100 N load at room temperature. Rectangular test pieces of strips 5 mm wide and 80 mm in length were cut from membrane, and they were subjected a tensile load applied at 5 mm min^−1^ until fracture.

Gas separation properties were measured in a constant volume/variable pressure apparatus using pure gases (He, O_2_, N_2_, CO_2_, and CH_4_) at 35 °C and an upstream pressure of 3 bar. The membrane was maintained inside the permeation cell under a high vacuum overnight to remove possible traces of humidity.

Permeability values were obtained from the evolution of pressure of the downstream side. The permeability coefficient, P (barrer), was determined from the slope of the pressure vs. time curve under steady-state conditions, where the relationship between permeate pressure and time is linear. Thus, the membrane permeability can be obtained from the following equation:(4)$$P\;\left(\mathrm{barrer}\right)=\frac{273.15}{76}\times \frac{\mathrm{Vl}}{{\pi r}^{2}{\mathrm{Tp}}_{o}}\times \frac{\mathrm{dp}}{\mathrm{dt}}\times 10^{10}$$
where V is the volume of the lower chamber in cm^3^, l is the thickness of the membrane in cm, T is the temperature in K, πr^2^ is the effective area of the membrane in cm^2^, p_0_ is the pressure in the upper chamber in bar, and dp/dt is the rate of the pressure rise under the steady state in mbar/s. The factors referenced are standard pressure and temperature conditions (76 cmHg and 273.15 K).

The ideal selectivity between two gases, A and B, is the ratio of the single gas permeability values (Equation (5)).
(5)$$\alpha_{A,B}=\frac{P_{A}}{P_{B}}$$

### 2.2. Synthesis of Copolyimides and Preparation of Polymer Films

Aromatic copolyimides with carboxyl groups were prepared by a two-step polycondensation reaction using the in situ silylation of the diamine method [61,62] to obtain the corresponding polyamic acid.

As an example, the synthetic methodology for a copolymer was:

In a three-necked flask equipped with mechanical stirring and N_2_ stream, 7.0 mmol of diamines (CF_3_TBAPB: 5.6 mmol, 3.000 g; DABA: 1.4 mmol, 0.213 g) were dissolved in 10 mL of DMAc. Once dissolved, the solution was cooled to 0 °C, and TMSCl (14 mmol, 1.8 mL) and anhydrous pyridine (14 mmol, 1.2 mL) were added. The solution temperature was maintained for 5 min and then raised to room temperature to ensure the silylation of the diamines. Next, the temperature reaction was lowered again to 0 °C, and 6FDA dianhydride (7.0 mmol, 3.100 g), DMAP (2.8 mmol, 0.342 g), and 5 mL of DMAc were added. The reaction was stirred for about 16 h at room temperature, giving a viscous polyamic acid solution.

Chemical cyclodehydration was carried out by adding an acetic anhydride/pyridine mixture (70 mmol/70 mmol). The mixture was stirred for 4 h at room temperature, heated to 40 °C for 1 h, and to 60 °C for a further 1 h. After cooling to room temperature, the polymer solution was precipitated in water/ethanol (2/1) and washed sequentially several times with water and water/ethanol (2/1). The polymers were dried at 150 °C in an oven under vacuum for 16 h.

The polymer films were prepared by dissolving the polymers in DMAc at 5–6% wt/vol. Subsequently, the solution was filtered using a 3.0 µm glass fiber Symta^®^ filter and spread on a glass plate placed over a pre-leveled heating plate. For solvent removal, the films were heated to 60 °C for 16 h, then the temperature was raised to 80 °C for 3 h more. Finally, the polymer film was heat-treated as follows: 150 °C/24 h/N_2_, 180 °C/30 min/N_2_, 200 °C/15 min/N_2_, 200 °C/15 min/vacuum, 250 °C/15 min/N_2_, and 280 °C/10 min/N_2_.

### 2.3. Crosslinking by Carboxylic Group Esterification

Crosslinking of the copolyimides was performed in solid-state as described in the literature, using 1,4-butanediol as the covalent crosslinking agent and *p*-toluenesulfonic acid as an acid catalyst [63].

Thus, a solution of copolymer (300 mg) in DMAc (10 mL) was prepared. Once dissolved, *p*-toluenesulfonic acid (3–5 mg per 1 g polymer) was added. After 30 min of stirring, 6 equivalents (per carboxyl group) of 1,4-butanediol were added and kept for 1 h more with stirring.

Subsequently, the membrane was obtained by casting. The solution was heated for 16 h at 60 °C and then raised to 80 °C for 3 h more. To achieve the maximum crosslinking of films, and to remove the unreacted compounds, the following heat treatment was carried out: 150 °C/24 h/N_2_ and gradually raising the temperature to 280 °C/10 min/N_2_.

Table 1 shows the designation used for the polymers, according to the proportion of diamines used.

Thus, the copolyimides were referred to as COPOL, the composition was specified by (1/X), where 1 is the proportion of DABA and X is the proportion corresponding to CF_3_TBAPB (X = 4 or X = 19). For the corresponding crosslinked films, COPOL-CCL (Covalent Crosslinking) will be used and the generic term HOMOPOL will be used to refer to the 6FDA-CF_3_TBAPB polyimide.

## 3. Results and Discussion

In previous works, our group synthesized aromatic polyimides derived from the CF_3_TBAPB diamine (Figure 2) and evaluated them as gas separation membranes, exhibiting excellent properties [64,65]. The main objective of this work was to improve the plasticization resistance of the CF_3_TBAPB-based polymers using copolymerization with DABA diamine, which allowed their chemical crosslinking through the carboxyl groups present in the polymeric materials.

In some cases, introducing carboxylic acid moiety into the polymeric structure improved the balance of properties. However, this improvement was not general because it was highly dependent on the structure of the other diamine comonomer [50].

The chemical structure of the synthesized copolyimides, as well as the proportions of the diamines used, are shown in Figure 3.

The proportions of DABA in the copolymer, and the amount of crosslinking agent used, 1,4-butanediol, were chosen based on the results obtained by Wind [46,64] and Staudt-Bickel [29,65].

Figure 4 shows the crosslinking reaction scheme of a copolyimide incorporating DABA with a substantial excess of 1,4-butanediol. In the first stage, initial monoesterification reactions occur, followed by subsequent crosslinking, by direct, or by transesterification, processes.

There is a probability that all the species indicated in Figure 4 (acid, monoester, and diester) coexist in the polymer material, although since the diol reacts in large stoichiometric excess, it can be assumed that the amount of free -COOH groups should be minimal. However, the most likely species will be the monoester groups, which could not have undergone crosslinking in the absence of optimum conditions. Another aspect to consider is that using a reagent excess makes it necessary to remove it from the material by prolonged washing or by heat treatments above the glass transition temperature, Tg.

### 3.1. Synthesis of Copolyimides, Preparation of Crosslinked Membranes

The synthesis of copolyimides was carried out in the same way as described for the homopolymer [64,65], adding a mixture of two diamines; DABA and CF_3_TBAPB, and using the in situ silylation method. The cyclization was carried out chemically because this method produced polymer films with better mechanical properties.

The preparation of the crosslinked films has been described above. Concerning the final temperature of the heat treatment, to achieve maximum crosslinking and minimum degradation, a final temperature of 280 °C was chosen.

### 3.2. Spectroscopic and Structural Characterization

#### 3.2.1. NMR Spectroscopy

Figure 5 and Figure 6 show the ^1^H-NMR spectra in CDCl_3_ of the synthesized copolyimides, COPOL-DABA(1/4) and COPOL-DABA(1/19) respectively. The proportions of DABA in the structure are relatively low, so no substantial changes were appreciated in the ^13^C-NMR spectra.

The protons in *ortho* to the carboxylic group of the DABA diamine were observed (signal marked with a blue ellipse in the NMR spectra). The protons of the acid groups were not observed under the conditions used in the experiment. These results agreed with others described previously for copolymers including the DABA monomer [50,66,67].

#### 3.2.2. ATR-FTIR Spectroscopy

Figure 7 shows the ATR-FTIR spectra of the films of the copolymers and the corresponding crosslinks, as well as that of the homopolymer. As can be seen, the patterns of the copolymers showed no significant differences concerning that of the homopolyimide. It should be noted that the modification was small, and therefore, the 1/4 and 1/19 ratios of DABA to CF_3_TBAPB make it difficult to appreciate the changes in their chemical structures. Thus, the stress vibration of the -OH groups, which appears as a broad band between 3000–3600 cm^–1^ when the DABA content is high, could not be observed [54].

When comparing the patterns of the copolyimides with those of their crosslinked counterparts, only small changes were observed in the region corresponding to the aliphatic C-H stress vibrations, around 2900 cm^–1^. However, it should be taken into account that this zone showed the bands of the *tert*-butyl groups of the CF_3_TBAPB diamine and those of the aliphatic chains of the crosslinking agent (1,4-butanediol), and they appeared to overlap. In conclusion, the ATR-FTIR technique was not suitable to characterize and evaluate the crosslinking of the copolymer membranes.

#### 3.2.3. Elemental Analysis

The elemental analyses of the copolyimides and the crosslinked films are shown in Table 2. Comparing the theoretical and experimental values, it can be seen that the values obtained for the copolyimides are close to the theoretical ones, although the differences observed in the case of the crosslinked copolyimides were somewhat larger.

#### 3.2.4. X-ray Diffraction (WAXS)

Figure 8 compares the diffraction patterns of copolymers films, crosslinked films, and the homopolymer. All of them showed an amorphous nature. At first glance, the major difference between the copolymer profiles with that of the homopolymer consisted of the decrease in intensity, or even in some cases the disappearance of the peak at low angles. This maximum peak could be associated with an intramolecular distance of 18.4 Å (θ = 4.8°), which could be related to the preference of the chains to adopt extended conformations. However, it can be also commented that the arrangement of the chains depends on many parameters, such as the imidization temperature, the heating rate, the type of substrate on which the film is prepared, and the thickness of the film, among others, which could justify the fact that sometimes a peak is detected or not.

Concerning crosslinked films, the COPOL-CCL(1/19) pattern seems to show a lower contribution to diffraction on the low-angle side of the amorphous halo. This fact could normally be associated with a higher packing of the chains. Chung et al. [46] commented that the maximum of the amorphous halo, which corresponded to the most probable intersegmental distance or spacing, was greater the higher the degree of crosslinking. They stated that when only mono-esterification reactions occurred, i.e., one -OH group of the diol remains unreacted with no crosslinking, the d-spacing was smaller. Taking these considerations into account, it could be thought that the lower contribution of large d-spacings to the amorphous halo of COPOL-CCL(1/19) was due to a high degree of mono-esterification.

#### 3.2.5. Solubility

Solubility is one of the fastest and most effective methods for testing crosslinking in polymeric materials. Crosslinking, by forming a network of infinite molecular weight, results in insolubility, although in membranes with small degrees of crosslinking, swelling, or partial solubility of the membranes is also often observed. Table 3 shows the solubilities in different solvents of the membranes studied.

Copolyimide membranes exhibited excellent solubilities in common solvents. It was observed that the COPOL-CCL(1/4) crosslinked membrane swelled when the solvent was heated, while COPOL-CCL(1/19) dissolved completely in all solvents tested, which seems to indicate that this membrane was either not crosslinked or had a very low degree of crosslinking.

Table 3 shows the values of gel fractions, Gf. It was found that COPOL-CCL(1/19) did not show any crosslinking process within the measurement error, while COPOL-CCL(1/4) showed a gel fraction around 95%. In addition, it was observed that the treated films retained dimensional stability when compared to the precursor films.

#### 3.2.6. Thermal Properties

Figure 9 presents the calorimetry curves (DSC) of the copolyimides and their crosslinked homologs. Table 4 lists the glass transition temperature values, Tg, for the copolyimides and their crosslinked homologs.

In general, Tg’s were higher than 260 °C. Moreover, the Tg of the copolyimides depended on the number of carboxylic groups; the higher the content, the higher the Tg. On the other hand, COPOL-DABA(1/4) and its crosslinked counterpart showed the same Tg. However, COPOL-CCL(1/19) presented a Tg 10 °C lower than the non-crosslinked copolyimide. This fact could be justified by the presence of unreacted crosslinking reagent, which was embedded inside the membrane and produced a plasticizing effect.

The comparison with the homopolymer 6FDA-CF_3_TBAPB (Tg = 270 °C) showed the expected behavior. Thus, the copolyimide with the larger proportion of DABA monomer presented the higher Tg, due to the rigidity conferred by this monomer and to the presence of hydrogen bonds between the carboxyl groups. For COPOL-DABA(1/19) no difference was appreciated, due to the small proportion of DABA present in the copolymer.

The most commonly accepted mechanism of carboxylic groups is that described by Kratochvil et al. [53,54]. This author assumed that the -COOH groups first react with each other to form a dianhydride, which subsequently decarboxylates at much higher temperatures, to form phenyl radicals that crosslink the polymer chains.

The decarboxylation process usually occurs between 350–400 °C (depending on the Tg of the polymer), although it has been reported that this crosslinking can occur at temperatures 50 °C below [13]. On the other hand, it has been published that it is possible to produce crosslinking in polyimides with -CF_3_ groups by treatment at elevated temperatures [55]. In this work, it was observed by using a TGA coupled to a mass spectrometer (MS), that one of the first by-products in the decomposition of this type of polyimide was HCF_3_. Thus, it is possible that at the same time as the acid groups are pyrolyzed, some CF_3_ groups decomposed with the result of a more crosslinked polyimide. Figure 10 shows the crosslinking mechanisms proposed in the literature and the possible crosslinking points [48,56,68].

Figure 11 shows the thermogravimetric curves of copolymer and crosslinked copolymer films. In addition, Table 4 shows the onset of degradation temperatures, Td, and carbonaceous residues (char yield, Rc (%)) at 800 °C under N_2_ (the Table also includes the values corresponding to the 6FDA-CF_3_TBAPB homopolymer).

COPOL-DABA(1/4) showed a well-defined weight loss step (Td of 375 °C), corresponding to the loss of acid groups (approximately 2% loss), while COPOL-DABA(1/19) showed only a small continued loss (0.5% between 350–400 °C) until the onset of generalized chain degradation, which was similar for both copolyimides to the Td of the homopolymer.

The crosslinked films also exhibited a small weight loss step before generalized chain degradation. In this case, the Tds were considerably lower than those observed for COPOL-DABA(1/4). In addition, COPOL-CCL(1/4) presented a small additional step at a temperature of 400 °C.

The weight losses in the first step can be associated with the loss of residual reagent, followed by the degradation of mono- or di-esterified chains. TGA showed that the COPOL-CCL(1/19) membrane had a higher amount of unreacted reagent. Furthermore, the solubility of this polymer indicated, as commented, that the degree of crosslinking was not high. Thus, many acid groups would be mono-esterified, and consequently, this polymer would have lower thermal stability [67].

The crosslinking that occurs in these materials, either by the covalent crosslinking carried out or by the thermal crosslinking derived from the thermolysis of the -COOH groups, gave rise to higher carbon residues than those observed for the 6FDA-CF_3_TBAPB homopolymer.

#### 3.2.7. Inherent Viscosity of Homopolymer and Copolyimides

The inherent viscosities of copolyimides are shown in Table 4.

Viscosities of copolyimides were not as high as that of the homopolymer. However, the good mechanical properties indicated that the polymers could be used as gas separation membranes at high pressures.

#### 3.2.8. Density and Fractional Free Volume (FFV)

In none of the publications on polyimide crosslinking, nor those dealing with DABA derivatives, FFV data were found. In the case of crosslinked membranes, the determination of FFV is quite cumbersome since the final structure of the polymer is not known, and therefore the determination of the molecular volume has a significant error. That is unless quantitative reactions are assumed, the determination of V_W_ volumes is not possible, and only the use of techniques that directly determine the FFV value, for example, using the PALS technique, would produce an accurate and rigorous analysis of the FFV (and its distribution) [69].

However, it was possible to determine the FFV values of the copolyimides (Table 5).

The trend in the density values obtained agreed with those described in the literature [70]. Thus, the copolyimides presented FFV values comparable (within the experimental error) to the homopolyimide. On the other hand, the crosslinked films presented higher densities than their counterparts without crosslinking (3% higher), although no significant conclusion could be drawn due to the solvent and crosslinking reagent embedded in the samples.

#### 3.2.9. Mechanical Properties

Table 6 shows the Young’s modulus, tensile strength, and elongation at break values of the copolyimide films, the crosslinked copolyimide films, and the homopolymer. The polymer films exhibited similar Young’s moduli, higher than 1.8 GPa. The crosslinked films showed higher Young’s moduli and lower tensile strength values than the copolyimides and the homopolymer. In addition, crosslinked films also showed lower elongation at break values, approximately half that of the non-crosslinked films.

### 3.3. Gas Separation Properties: Permeability, Selectivity, and Plasticization Study

#### 3.3.1. Permeability and Selectivity of Membranes

Table 7 shows the gas separation results obtained for the homopolyimide, DABA-derived copolyimides, and covalently crosslinked polymers. These data were compared with the results presented previously for the 6FDA-CF_3_TBAPB(C) homopolymer.

A decrease in permeability to all gases when comparing these copolyimides with the homopolymer was observed. Thus, the substitution, even in very small proportions, of the diamine CF_3_TBAPB by the diamine DABA, resulted in a decrease in permeability for both copolyimides. The selectivity underwent an important improvement that could be associated with a greater intrinsic rigidity of the chains, introduced by the DABA diamine, and with the existence of polar groups, which can interact with certain gases. However, it could be seen that there seems to be an inverse relationship between the amount of COOH groups present in the polymeric matrix and their effect on permeability (both for the O_2_/N_2_ gas pair and for the CO_2_/CH_4_ one).

Thus, it was observed that the membrane with a lower amount of DABA in its structure presented a more pronounced decrease in permeability:P_HOMOPOL_ > P_COPOL-DABA(1/4)_ > P_COPOL-DABA(1/19)_.

With the introduction of 20% DABA in the main chain of COPOL-DABA(1/4), the permeability for both O_2_ and CO_2_ was reduced by 20% relative to the homopolymer. However, with the introduction of only 5% DABA in the main chain of COPOL-DABA(1/19), the permeability was reduced by more than 30% for CO_2_ and 50% for O_2_. This behavior is strange and we could not find an explanation for this anomalous fact.

In the case of the crosslinked membranes, a decrease in permeability values was observed when compared with the non-crosslinked counterparts, which did not translate into an improvement in selectivity. The material with the lowest content of carboxyl groups had the lowest permeabilities, which could be due to the higher amount of residual reagent present inside the membrane. No differences in O_2_/N_2_ and CO_2_/CH_4_ selectivity values were observed between the crosslinked membranes and the reference copolyimides.

The degree of productivity of the membranes was determined, by plotting these systems in graphs of permeability vs. selectivity where Robeson limits [71,72] were included. Figure 12 and Figure 13 plot selectivity (α O_2_/N_2_ and α CO_2_/CH_4_) vs. permeability to the most permeable gas of both mixtures (PO_2_ and PCO_2_, respectively).

Figure 13 shows the improvements in selectivity, for the copolyimides and crosslinked copolyimides, compared to the homopolymer for the O_2_/N_2_ gas pair. The permeabilities of all of them were comparable, except for COPOL-CCL(1/19), which was lower.

Therefore, the objective of improving the gas separation properties of the homopolymer (6FDA-CF_3_TBAPB polyimide) in this series of copolyimides has been achieved. It could be concluded that the introduction of -COOH groups in the polymeric structure in the ratio used (1/4) and (1/19) lead to an enhancement in the permeability/selectivity balance for air purification processes since the loss in permeability was not very pronounced in comparison with the selectivity improvement.

In the case of the CO_2_/CH_4_ gas pair (Figure 14), a similar behavior was observed. In other words, an increase in selectivity was observed in all cases having a slight loss of permeability.

In conclusion, the excellent results obtained in the gas separation measurements confirmed that the COPOL-DABA(1/4), COPOL-DABA(1/19), and COPOL-CCL(1/4) membranes showed good permeation properties and they were more selective than the 6FDA-CF_3_TBAPB polyimide. Therefore, it could be stated that using a simple, cheap, and easy-to-perform copolymerization with the DABA monomer, it was possible to improve the separation properties for several gas mixtures. However, it should be noted that in the case of COPOL-DABA(1/19), having a very low DABA content, it was not possible to obtain the desired degree of crosslinking.

#### 3.3.2. Plasticization Study

Finally, a study was carried out to determine the tendency of these membranes to undergo plasticization processes. This study consisted of monitoring the changes in permeability in each of the membranes when CO_2_ passed through them at different pressures. The plasticization pressure, Pp, or pressure at which the increase in permeability begins to occur if it exists, could be obtained [21,73].

Figure 14 shows the CO_2_ permeability data at different pressures. The CO_2_ pressures used were: 3, 5, 7, 9, 11, 13, 15, 20, 25, and 30 bar.

The behavior shown by the membranes agreed with that described in the literature for copolyimides derived from 6FDA and DABA. In the copolyimide membranes with free -COOH groups, COPOL-DABA(1/4) and COPOL-DABA(1/19), plasticization pressures were observed in the zone between 15 and 20 bar. However, the increase in permeability that occurs from this pressure was much less pronounced than that observed for most of the polyimides reported in the literature, where permeability generally increases more sharply above that pressure value. Thus, it can be said that the copolyimides derived from CF_3_TBAPB, and having DABA as comonomer, presented an improved plasticization resistance when compared with other polyimides derived from the 6FDA dianhydride.

For the COPOL-CCL(1/4) crosslinked membrane, it was observed, when compared with the precursor copolyimide, a considerable decrease in the curvature of the graph, so, the increase in permeability from the plasticization pressure point was much lower.

In the case of COPOL-CCL(1/19), a slightly lower plasticization pressure was observed than that of the starting membrane COPOL-DABA(1/19), which seems to confirm that this membrane is not crosslinked and contains a large amount of crosslinking reagent. This statement is in agreement with the previously cited work of Hess et al. [63].

As can be seen in Figure 14, COPOL-CCL(1/4) presented a low plasticization tendency.

## 4. Conclusions

An adequate design has permitted us to obtain a new family of aromatic polyimides derived from a mixture of a rigid diamine 1,4-bis(4-amino-2-trifluoromethylphenoxy)-2,5-di-*tert*-butylbenzene (CF_3_TBAPB) and another diamine having carboxylic groups, 3,5-diamino benzoic acid (DABA).

Thanks to the carboxylic group present in the structure, the polymers were chemically crosslinked by employing a difunctional diol. The crosslinked films showed good mechanical properties, and glass transition temperatures were comparable to the precursor copolyimides.

The chemical crosslinking did not impair the gas separation properties of the crosslinked polyimides, showing similar permeabilities and a slightly better permselectivity. In addition, one of the crosslinked copolyimides showed a low tendency to be plasticized.

In this work, important differences have been observed between the copolyimide membrane having free acid groups, its crosslinked counterpart, and the non-crosslinked one, which is a membrane with mostly mono-esterification moieties. It was observed that the COPOL-CCL(1/4) presented a low tendency to plasticize together with good permeability values. This result allows us to consider that the approach of incorporating small percentages of DABA, as a comonomer, in homopolymers having high free volume fractions, is a valid approach to obtaining materials with excellent gas separation properties.

Finally, it should be stated that to know the real applicability of these materials in industrial-level gas separation, it seems necessary to know their resistance to plasticization in long-term separation processes and, above all, to study their behavior in real gas mixtures at high pressures and temperatures.

## Figures and Tables

**Figure 1 polymers-14-05517-f001:**
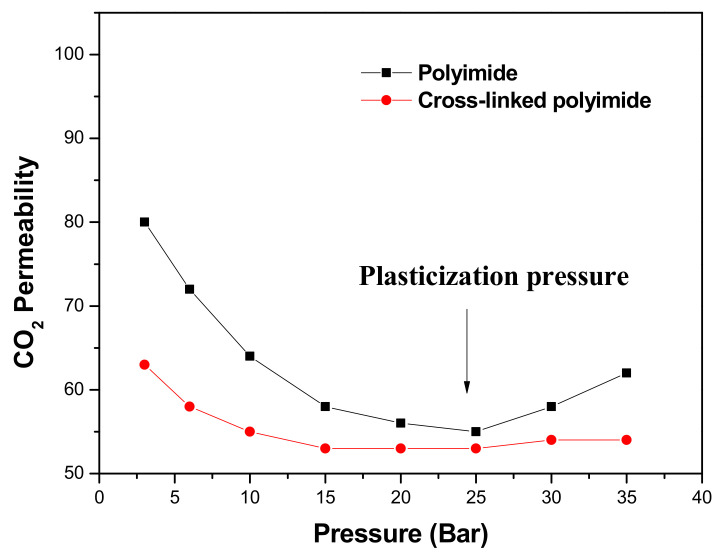
Plasticization process in a linear and a crosslinked polyimide [5].

**Figure 2 polymers-14-05517-f002:**
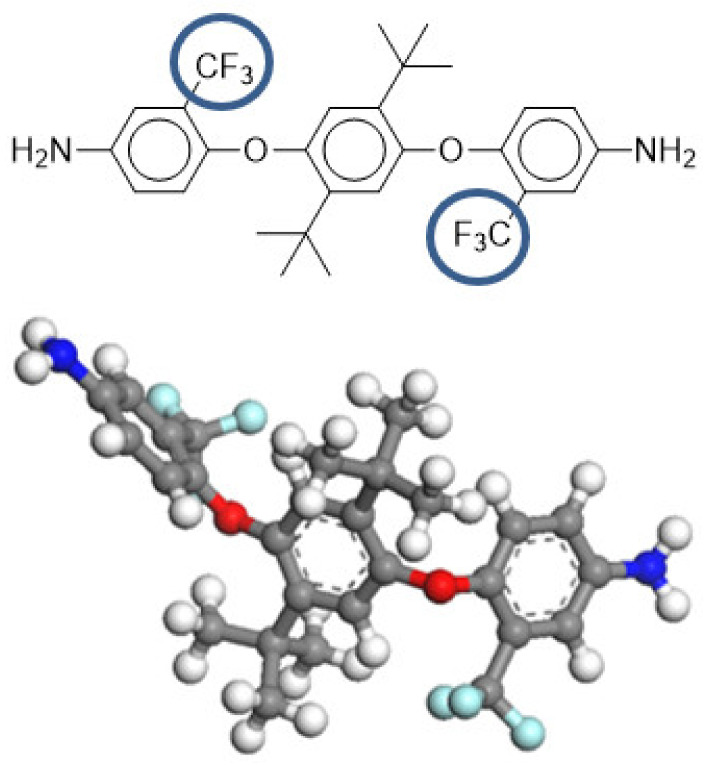
Structure of 4-bis(4-amino-2-trifluoromethylphenoxy)2,5-di-*tert*-butylbencene (CF_3_TBAPB).

**Figure 3 polymers-14-05517-f003:**
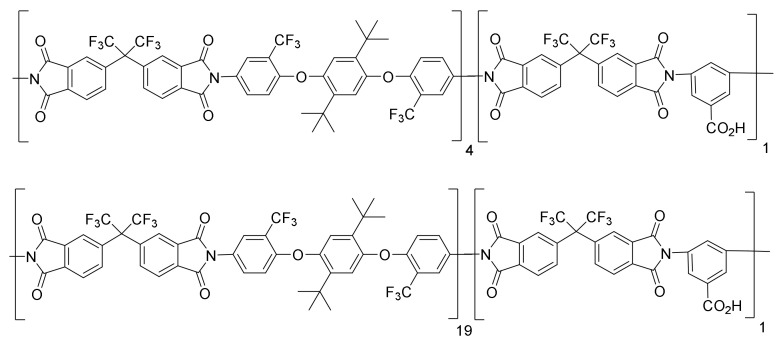
Chemical structure of copolymers 6FDA-[DABA-CF_3_TBAPB (1/4)] and 6FDA-[DABA-CF_3_TBAPB (1/19)].

**Figure 4 polymers-14-05517-f004:**
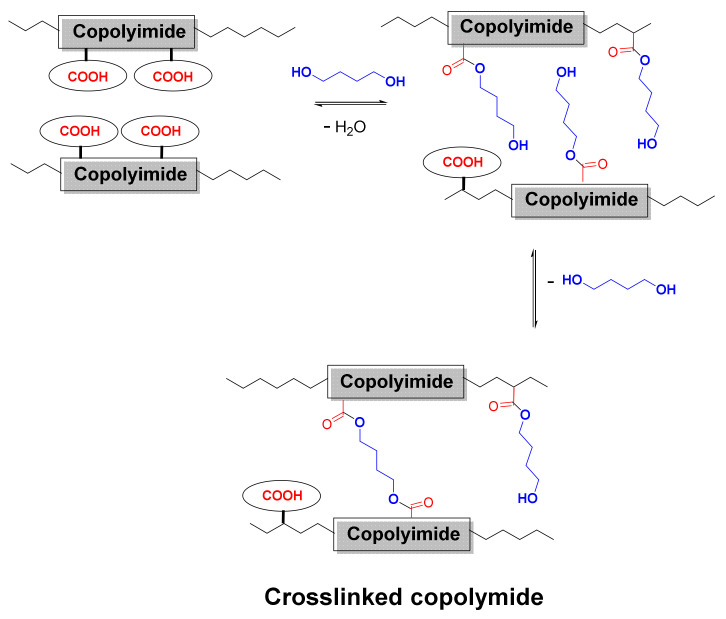
Scheme of covalent crosslinking reaction in a copolyimide incorporating DABA.

**Figure 5 polymers-14-05517-f005:**
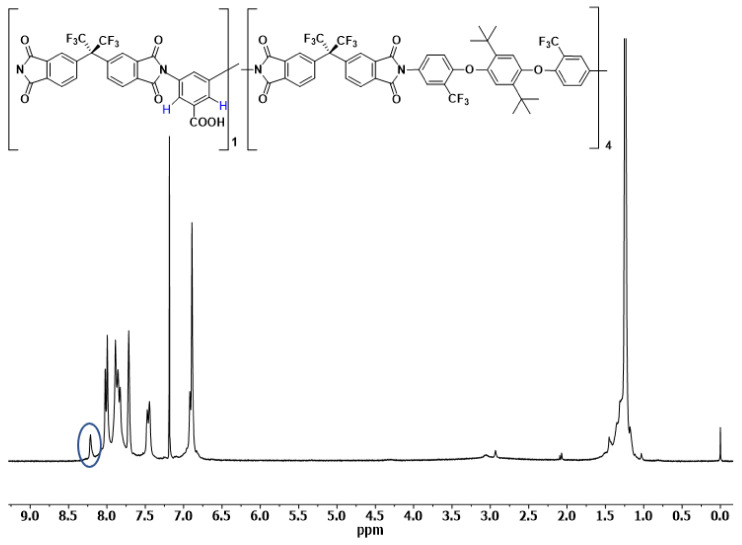
^1^H-NMR spectrum of COPOL-DABA(1/4).

**Figure 6 polymers-14-05517-f006:**
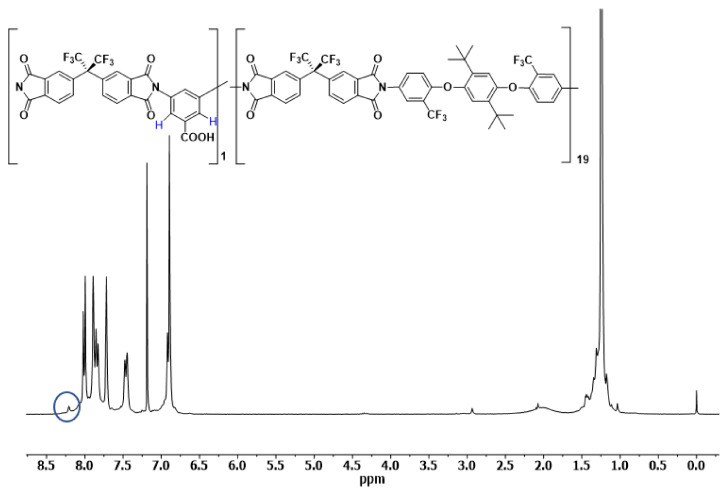
^1^H-NMR spectrum of COPOL-DABA(1/19).

**Figure 7 polymers-14-05517-f007:**
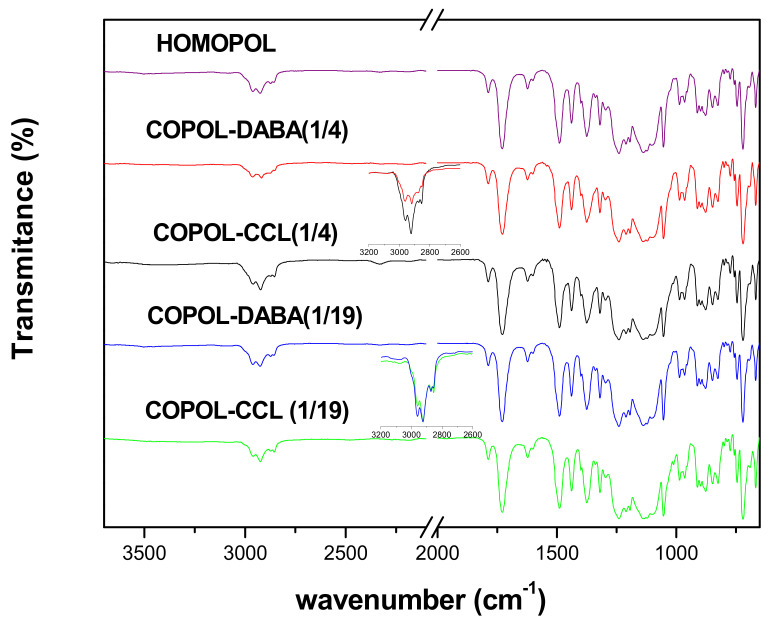
ATR-FTIR spectra for the homopolymer, the copolyimides, and the crosslinked homologs. The insets in the graphs show the broadened region around 2900 cm^–1^.

**Figure 8 polymers-14-05517-f008:**
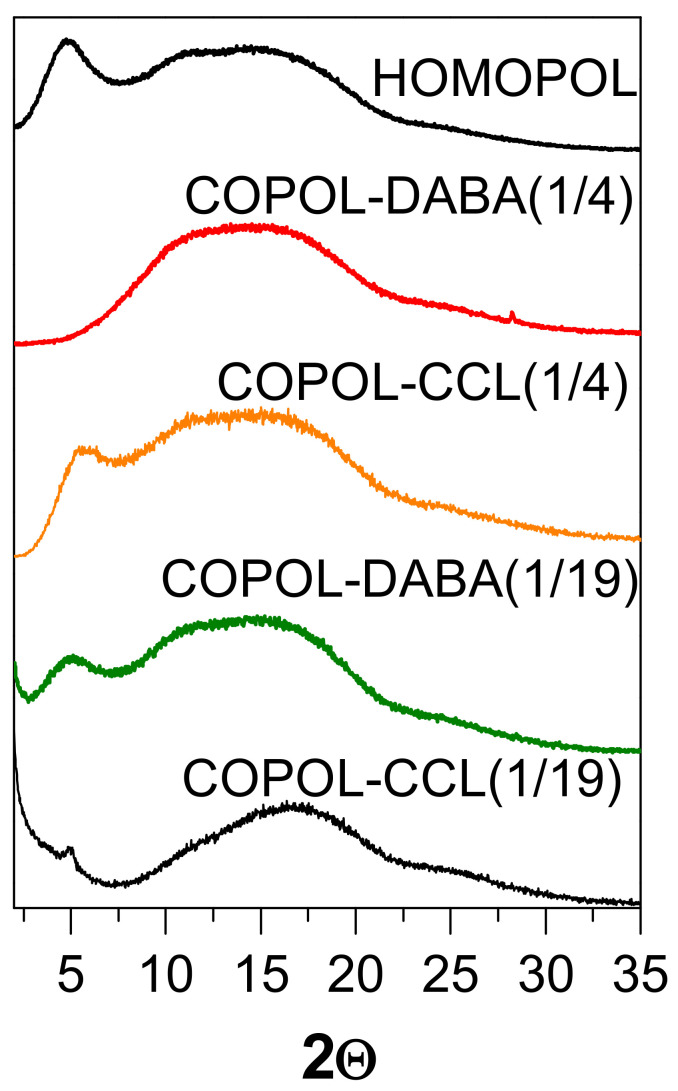
WAXS diffractograms of the homopolymer, DABA-derived copolyimides, and their corresponding crosslinked homologs.

**Figure 9 polymers-14-05517-f009:**
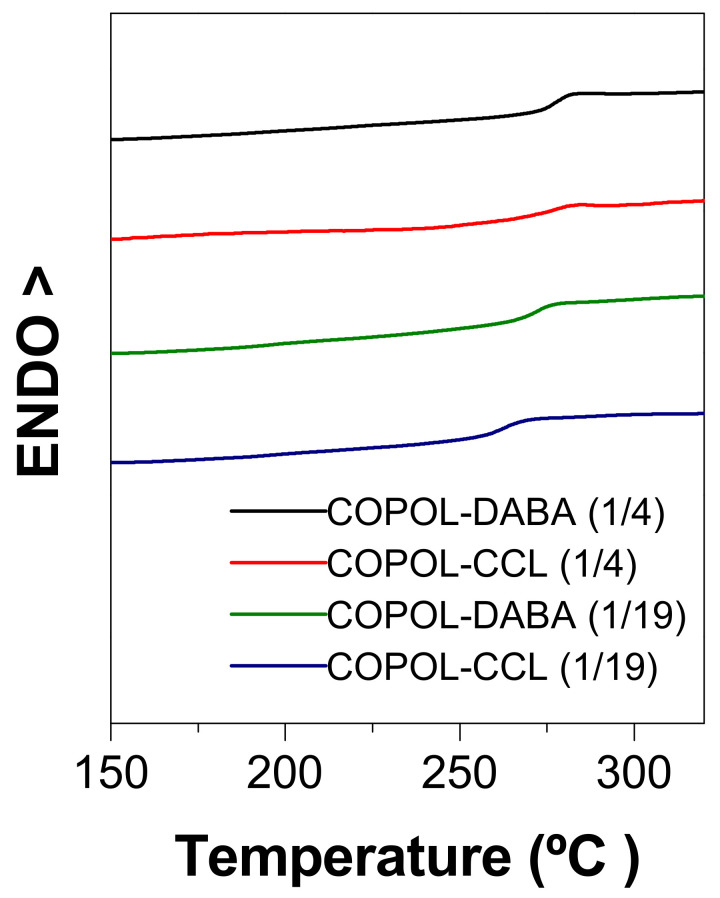
DSC of copolyimides and their corresponding crosslinked films.

**Figure 10 polymers-14-05517-f010:**
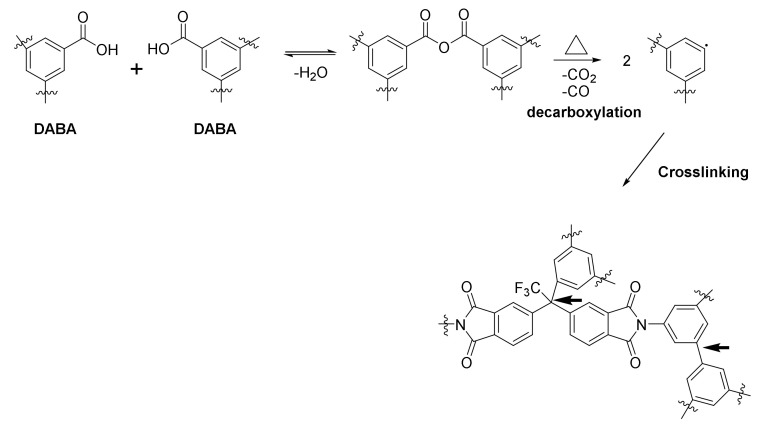
Crosslinking mechanism for copolymers incorporating DABA monomer and CF_3_ groups [53] (bold arrows represent the plausible crosslinking points).

**Figure 11 polymers-14-05517-f011:**
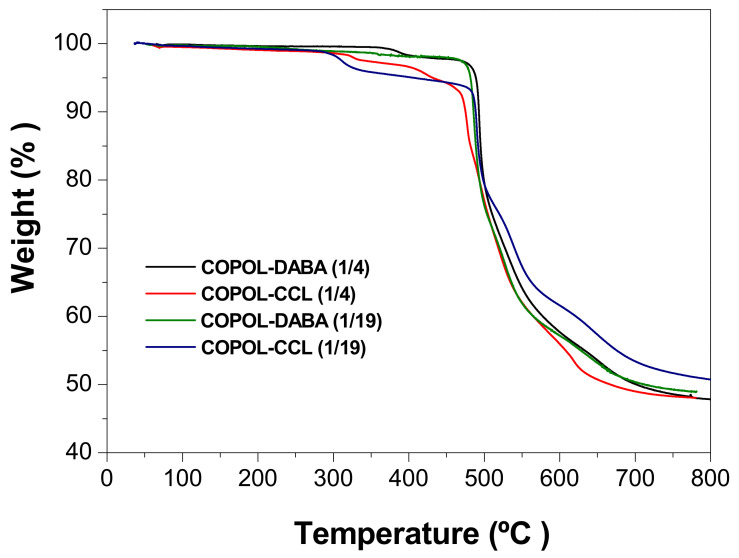
Thermograms of copolyimides and their crosslinked counterparts.

**Figure 12 polymers-14-05517-f012:**
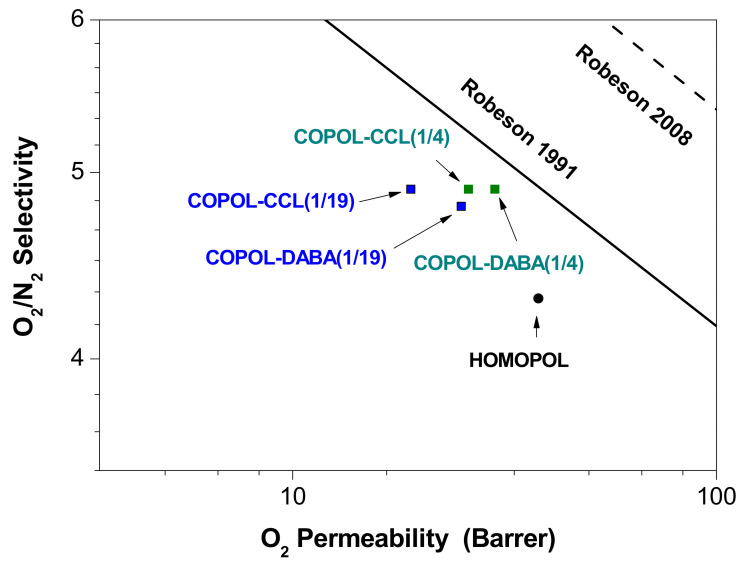
O_2_/N_2_ selectivity vs. O_2_ permeability [71,72].

**Figure 13 polymers-14-05517-f013:**
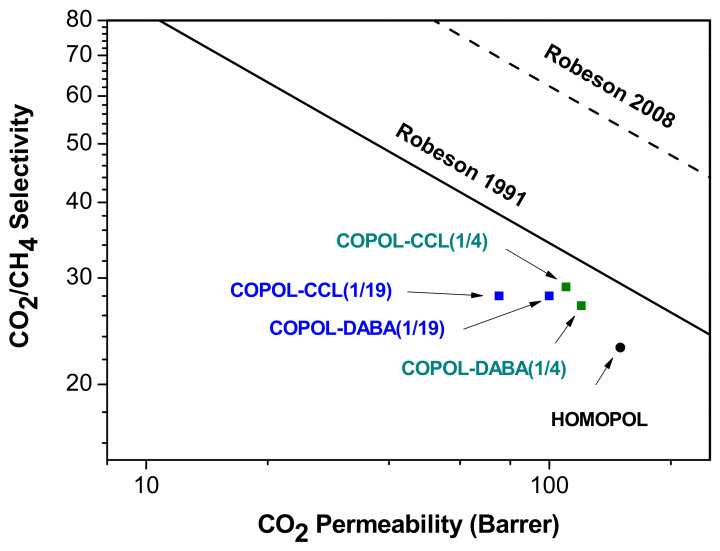
CO_2_/CH_4_ selectivity vs. CO_2_ permeability [71,72].

**Figure 14 polymers-14-05517-f014:**
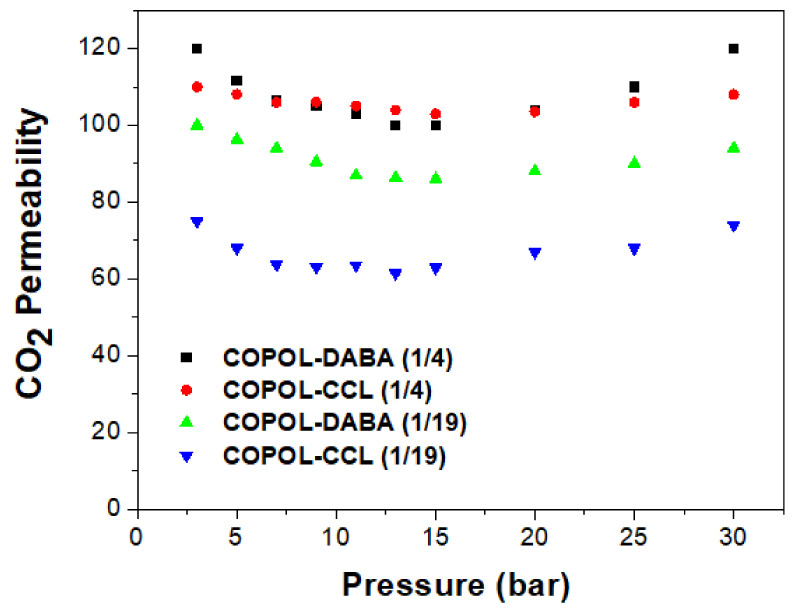
CO_2_ permeability vs. pressure.

**Table 1 polymers-14-05517-t001:** Acronyms of polymers.

Polymer	Acronym
6FDA-CF_3_TBAPB	HOMOPOL
6FDA-[DABA-CF_3_TBAPB (1/4)]	COPOL-DABA(1/4)
6FDA-[DABA-CF_3_TBAPB (1/4)] *	COPOL-CCL(1/4)
6FDA-[DABA-CF_3_TBAPB (1/19)]	COPOL-DABA(1/19)
6FDA-[DABA-CF_3_TBAPB (1/19)] *	COPOL-CCL(1/19)

* Solid-state covalently crosslinked films with 1,4-butanediol.

**Table 2 polymers-14-05517-t002:** Elemental analysis of copolyimides and their crosslinked counterparts.

Polymer	%C Theoretical/Found	%H Theoretical/Found	%N Theoretical/Found
COPOL-DABA(1/4)	58.8/58.7	3.1/3.3	3.4/3.1
COPOL-CCL-(1/4)	59.1 */58.5 58.9 **	3.2 */3.0 3.2 **	3.3 */3.0 3.3 **
COPOL-DABA(1/19)	59.3/59.2	3.3/3.5	3.1/2.9
COPOL-CCL-(1/19)	59.4 */58.3 59.4 **	3.3 */3.3 3.4 **	3.0 */2.8 3.03 **

* Theoretical % assuming 100% crosslinking. ** Theoretical % assuming -OH-terminal modifications (monoester entities).

**Table 3 polymers-14-05517-t003:** Solubility of copolyimides and gel fraction.

Polymer	CHCl_3_	THF	DMAc	NMP	m-Cresol	Gel Fraction, Gf (%)
COPOL-DABA(1/4)	+ +	+ +	+ +	+ +	+ +	0
COPOL-CCL(1/4)	+ -	+ -	+ -	+ -	+-	>95
COPOL-DABA(1/19)	+ +	+ +	+ +	+ +	+ +	0
COPOL-CCL(1/19)	+	+	+	+	+	<5

Legend applied: + + cold soluble, + hot soluble, +- partially soluble in hot, —insoluble.

**Table 4 polymers-14-05517-t004:** Glass transition temperatures (Tg), degradation temperatures (Td), and char yield (Rc) at 800 °C, under nitrogen atmosphere, of homopolymer and copolyimides, and crosslinked polyimides. Inherent viscosity (ηinh) values of homopolyimide and copolyimides.

Polymer	Tg (°C)	Td (°C)	Rc (%)	ηinh (dL/g)
HOMOPOL	270	490	40	1.09
COPOL-DABA(1/4)	280	375/490	47	0.69
COPOL-CCL(1/4)	280	315/400/480	50	-
COPOL-DABA(1/19)	270	480	49	0.66
COPOL-CCL(1/19)	260	295/490	51	-

**Table 5 polymers-14-05517-t005:** Film densities of homopolyimide, copolyimides, and crosslinked polyimides. Fractional free volumes (FFV) of homopolyimide and copolyimides.

Polymer	ρ (g/cm^3^)	FFV
HOMOPOL	1.291	0.238
COPOL-DABA(1/4)	1.301	0.235
COPOL-CCL(1/4)	1.340	-
COPOL-DABA(1/19)	1.295	0.233
COPOL-CCL(1/19)	1.334	-

**Table 6 polymers-14-05517-t006:** Mechanical properties of polymers.

Polymer	Young Modulus (GPa)	Tensile Strength (MPa)	Elongation at Break (%)
HOMOPOL	1.6 ± 0.1	91 ± 10	9 ± 2
COPOL-DABA(1/4)	1.7 ± 0.2	96 ± 18	9 ± 1
COPOL-CCL-(1/4)	1.9 ± 0.1	75 ± 12	5 ± 1
COPOLDABA(1/19)	1.9 ± 0.1	105 ± 5	10 ± 2
COPOL-CCL-(1/19)	2.1 ± 0.1	88 ± 4	5.7 ± 0.2

**Table 7 polymers-14-05517-t007:** Permeability (P) and ideal selectivity (α) coefficients for linear and crosslinked copolyimides (measurement conditions: 3 bar at 35 °C).

Polymer	He	N_2_	O_2_	CH_4_	CO_2_	αO_2_/N_2_	αCO_2_/CH_4_
HOMOPOL	230	8.7	38	6.4	150	4.3	23
COPOL-DABA(1/4)	200	6.1	30	4.5	120	4.9	27
COPOL-CCL(1/4)	180	5.4	26	3.8	110	4.9	29
COPOL-DABA(1/19)	170	5.1	25	3.7	100	4.8	28
COPOL-CCL(1/19)	140	3.8	19	2.7	75	4.9	28

Permeability in Barrer. 1 Barrer = 10^–10^ cm^3^ (STP) cm/cm^2^ s cmHg.

## Data Availability

Not applicable.

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
