# Peer review of "Aromatic Polyimide Membranes with tert-Butyl and Carboxylic Side Groups for Gas Separation Applications—Covalent Crosslinking Study"

_polymers, 2022, doi:10.3390/polym14245517_

Round 1

Reviewer 1 Report

The report is titled "Aromatic polyimide membranes with tert-butyl and carboxylic side groups for gas separation applications. Covalent crosslinking study" and nicely fits the journal. There are some minor corrections that the authors could consider.

The introduction is too long and tedious; from page 3, the last paragraph starting with "Aromatic... to the last paragraph of the introduction on the fifth page" should transfer to the result and discussion parts.

The authors have not measured the aging effect of the prepared membrane, which is an important parameter to this work, they could include.

There have been more seminal works published in recent years; the authors could update their references with recently published works.

Reviewer 2 Report

1. The abstract should state briefly the purpose of the research, the principal results and major conclusions. The authors should rewrite the abstract for these standards.

2. The conclusion has to describe the achieved results. Generally conclusion and abstract have to be connected and have to answer the same questions.

3. Introduction describes some research about crosslinking studies. However, the method of how the crosslinked was measured in the polymer membrane in the current research wasn't present. Maine structural parameters of the polymer network, like crosslinking degree, have to be shown.

4. The following references focusing on crosslinking studies are suggested to be cited DOI: 10.15587/1729-4061.2020.216745

5. In the section of introduction and analysis, the following references focusing on gas removal are suggested to be cited

DOI: 10.11159/iceptp22.185

DOI: 10.1016/j.molliq.2022.120287

DOI: 10.3390/membranes11020097
